# Modes of Interactions with DNA/HSA Biomolecules and Comparative Cytotoxic Studies of Newly Synthesized Mononuclear Zinc(II) and Heteronuclear Platinum(II)/Zinc(II) Complexes toward Colorectal Cancer Cells

**DOI:** 10.3390/ijms25053027

**Published:** 2024-03-06

**Authors:** Samir Vučelj, Rušid Hasić, Darko Ašanin, Biljana Šmit, Angelina Caković, Jovana Bogojeski, Marina Ćendić Serafinović, Bojana Simović Marković, Bojan Stojanović, Sladjana Pavlović, Isidora Stanisavljević, Irfan Ćorović, Milica Dimitrijević Stojanović, Ivan Jovanović, Tanja V. Soldatović, Bojana Stojanović

**Affiliations:** 1Faculty of Medical Sciences, Center for Molecular Medicine and Stem Cell Research, University of Kragujevac, S. Markovića 69, 34000 Kragujevac, Serbia; vucelj.samir@hotmail.com (S.V.); bojana.simovic@gmail.com (B.S.M.); bojan.stojanovic01@gmail.com (B.S.); sladjadile@gmail.com (S.P.); isidorastanisavljevic97@gmail.com (I.S.); ira.corovic@gmail.com (I.Ć.); milicadimitrijevic@yahoo.com (M.D.S.); ivanjovanovic77@gmail.com (I.J.); bojana.stojanovic04@gmail.com (B.S.); 2General Hospital of Novi Pazar, Department of Internal Medicine, Generala Živkovića 1, 36300 Novi Pazar, Serbia; 3Department of Natural-Mathematical Sciences, State University of Novi Pazar, Vuka Karadžića 9, 36300 Novi Pazar, Serbia; rusid.np.chem@gmail.com; 4Institute for Information Technologies, University of Kragujevac, Jovana Cvijića bb, 34000 Kragujevac, Serbia; darko.asanin@uni.kg.ac.rs (D.A.); biljana.smit@uni.kg.ac.rs (B.Š.); 5Faculty of Science, University of Kragujevac, Radoja Domanovića 12, 34000 Kragujevac, Serbia; angelina.petrovic@pmf.kg.ac.rs (A.C.); jovana.bogojeski@pmf.kg.ac.rs (J.B.); marina.cendic@pmf.kg.ac.rs (M.Ć.S.); 6Faculty of Medical Sciences, Department of Surgery, University of Kragujevac, S. Markovića 69, 34000 Kragujevac, Serbia; 7Faculty of Medical Sciences, Department of Pathology, University of Kragujevac, S. Markovića 69, 34000 Kragujevac, Serbia; 8Faculty of Medical Sciences, Department of Pathophysiology, University of Kragujevac, S. Markovića 69, 34000 Kragujevac, Serbia

**Keywords:** heteronuclear complexes, 4,4′,4″-tri-*tert*-butyl-2,2′:6′,2″-terpyridine, structure–reactivity correlation, cytotoxic activity, colorectal cancers

## Abstract

A series of mono- and heteronuclear platinum(II) and zinc(II) complexes with 4,4′,4″-tri-*tert*-butyl-2,2′:6′,2″-terpyridine ligand were synthesized and characterized. The DNA and protein binding properties of [ZnCl_2_(terpy*^t^*^Bu^)] (**C1**), [{*cis*-PtCl(NH_3_)_2_(*μ*-pyrazine)ZnCl(terpy*^t^*^Bu^)}](ClO_4_)_2_ (**C2**), [{*trans*-PtCl(NH_3_)_2_(*μ*-pyrazine)ZnCl(terpy*^t^*^Bu^)}](ClO_4_)_2_ (**C3**), [{*cis*-PtCl(NH_3_)_2_(*μ*-4,4′-bipyridyl)ZnCl(terpy*^t^*^Bu^)}](CIO_4_)_2_ (**C4**) and [{*trans*-PtCl(NH_3_)_2_(*μ*-4,4′-bipyridyl)ZnCl(terpy*^t^*^Bu^)}](CIO_4_)_2_ (**C5**) (where terpy*^t^*^Bu^ = 4,4′,4″-tri-*tert*-butyl-2,2′:6′,2″-terpyridine), were investigated by electronic absorption, fluorescence spectroscopic, and molecular docking methods. Complexes featuring transplatin exhibited lower *K*_b_ and *K*_sv_ constant values compared to cisplatin analogs. The lowest *K*_sv_ value belonged to complex **C1,** while **C4** exhibited the highest. Molecular docking studies reveal that the binding of complex **C1** to DNA is due to van der Waals forces, while that of **C2**–**C5** is due to conventional hydrogen bonds and van der Waals forces. The tested complexes exhibited variable cytotoxicity toward mouse colorectal carcinoma (CT26), human colorectal carcinoma (HCT116 and SW480), and non-cancerous mouse mesenchymal stem cells (mMSC). Particularly, the mononuclear **C1** complex showed pronounced selectivity toward cancer cells over non-cancerous mMSC. The **C1** complex notably induced apoptosis in CT26 cells, effectively arrested the cell cycle in the G0/G1 phase, and selectively down-regulated Cyclin D.

## 1. Introduction

The design of novel heteronuclear complexes as potential anticancer agents has become a new trend to enhance the anticancer properties of single metallodrugs [1,2]. Although platinum-based drugs show outstanding antitumor activity, their effectiveness is limited by some unwanted side effects such as resistance and toxicity, manifested in nephrotoxicity, neurotoxicity, ototoxicity, cardiotoxicity, etc. [3,4,5,6]. Incorporating two different metal ions, where one is a biologically active metal, in the same molecule may improve antitumor activity. This improvement can result from selective interactions of metal ions with biomolecules or the ameliorative effect of chemical–physical properties of the resulting heteronuclear complexes (synergism) [7].

The mechanism of action of platinum-based drugs is very well known, and antitumor activity is explained by interactions between the platinum(II) complexes and DNA, mostly via N7 atoms of guanine, while negative side effects are explained by interactions with biomolecules that contain sulfur [3,4,5]. Zinc(II) ion as a borderline Lewis acid has an affinity to coordinate both nitrogen or oxygen donor atoms in biomolecules [7,8,9]. Our recent studies indicate that heteronuclear complexes with adopted general formula *cis*- or *trans*-Pt-L-Zn are more reactive toward DNA constituents than sulfur-containing biomolecules. These complexes have demonstrated significantly higher cytotoxicity compared to cisplatin [10,11]. The increased cytotoxicity may be attributed to potential interference with multiple intracellular processes and structures that incorporate the zinc(II) ion. The molecular mechanisms governing the response of the colorectal cancer cell line HCT116 to the cytotoxic action of these complexes appear to be complex, involving intricate interplays between various signaling pathways [10,11,12].

The newly designed heteronuclear gold(I)/platinum(II) complexes with bipyridylamine–phosphine ligands showed marked antiproliferative effects in the range of cisplatin or higher [13]. Cytotoxicity assay shows that ruthenium(II)/rhenium(I) also exhibits higher anticancer activity than cisplatin. The complexes induce apoptosis through the regulation of the cell cycle, depolarization of mitochondrial membrane potential elevation of intracellular reactive oxygen species, and caspase cascade [14]. The new platinum(IV)/ruthenium(II) complexes, which combine the cytotoxic properties of cisplatin with the antimetastatic characteristics of ruthenium–arene complexes, were found to exhibit higher cytotoxicity compared to cisplatin. They showed low micromolar, sub-micromolar, and nanomolar cytotoxicity in the majority of tested human cancer cells. Additionally, these heteronuclear complexes proved effective even in cisplatin-resistant cancer cells, specifically A2780cisR (ovarian carcinoma) and A549cisR (lung adenocarcinoma) [15,16]. The design and development of heterometallic complexes as anticancer drugs involve a multidisciplinary approach, combining knowledge from chemistry, biology, and medicine.

In summary, novel heteronuclear complexes are designed according to a strategy aimed at directing them toward cancer-specific biomarkers or biological targets overexpressed in cancer cells. The goal is to attain superior selectivity compared to platinum-based drugs while minimizing side effects [1,2].

The main goal of this study was the synthesis and characterization of mononuclear and four novel heteronuclear complexes with a substituted terpyridine ligand, as well as the determination of DNA and protein binding properties in order to achieve higher cytotoxic properties. The structures of investigated [ZnCl_2_(terpy*^t^*^Bu^)] (**C1**), [{*cis*-PtCl(NH_3_)_2_(*μ*-pyrazine)ZnCl(terpy*^t^*^Bu^)}](ClO_4_)_2_ (**C2**), [{*trans*-PtCl(NH_3_)_2_(*μ*-pyrazine)ZnCl(terpy*^t^*^Bu^)}](ClO_4_)_2_ (**C3**), [{*cis*-PtCl(NH_3_)_2_(*μ*-4,4′-bipyridyl)ZnCl(terpy*^t^*^Bu^)}](CIO_4_)_2_ (**C4**), and [{*trans*-PtCl(NH_3_)_2_(*μ*-4,4′-bipyridyl)ZnCl(terpy*^t^*^Bu^)}](CIO_4_)_2_ (**C5**) (where terpy*^t^*^Bu^ = 4,4′,4″-tri-*tert*-butyl-2,2′:6′,2″-terpyridine) are provided in Figure 1.

## 2. Results and Discussion

### 2.1. Synthesis and Characterization

The novel **C1**–**C5** complexes were synthesized and characterized by elemental analyses, UV–Vis, IR, ^1^H NMR, and ESI-MS spectroscopy. The general pathway of the synthesis of novel heteronuclear **C2**–**C5** complexes is presented in Figure 1. Aqua species were obtained during synthesis by adding AgClO_4_, forasmuch as perchlorate anions have a low binding ability to both metals. Detailed results from spectroscopic analysis are listed in the Experimental section and in Appendix A.

The ^1^H NMR spectra of the complexes have shown multiplets formed by overlapped signals of 3,5 and 3″,5″ protons in the range of 9.25–8.84 ppm. Singlets of 3,3′,3″ and 5′ protons from the middle pyridine ring were in span 7.99–7.07 ppm. Singlets of tri-*tert*-butyl groups appeared in the range 1.48–1.23 ppm, while singlets of protons from –NH_3_ groups were between 4.75 and 1.46 ppm. The signals of bridging pyrazine ligands were at 8.99–8.74 ppm, while for 4,4′-bipyridyl ligands, they overlapped with signals 2,2″,3,3″ from terpyridine and appeared in the region 8.98–8.26. The only difference between ^1^H NMR spectra of **C4** and **C5** complexes was noticed in the position of signals corresponding to protons from -NH_3_ groups, which is a consequence of geometry around the platinum(II) center. In the **C4** complex, two –NH_3_ groups are in the *cis* position, which has a downfield shift of singlet compared to *trans* **C5** isomers.

IR spectra of investigated complexes showed the bands attributed to the -NH_3_ group in the range 3436–3293 cm^−1^, bands specified for aromatic ν(C=C) and ν(C=N) stretching vibrations in the region 1619–1391 cm^−1^. Broad bands of ν(C=N) at 1659, 1619, 1617, and 1607 cm^−1^ correspond to complexes with bridging pyrazine or 4,4′-bipyridyl. The bands in the region 627–615 cm^−1^ were attributed to ν(Zn–N), while bands at 821, 820, 866, 819, and 777 correspond to ν(C=C).

The UV–Vis spectra of the complexes were recorded in water and PBS buffer, and no differences were observed. The intensive bands for the **C2**–**C5** complexes were displayed in 274, 284, 306, 316, and 328 nm (**C2**) and 274, 284, 318, and 330 nm (**C3**), while for the **C4** and **C5,** one λ_max_ was at 282 nm and 278 nm, respectively. The results indicated different electronic communications between two metal centers and correspond to ligand-to-metal charge transfer transition.

Mass spectra of the **C1**–**C5** complexes were recorded in a positive ion mode. The signal at *m*/*z* 577.99 matches to [M + K]^+^ adduct for **C1** complex. Molecular ions detected at *m*/*z* 344.02 (±0.20) and 488.04 (±0.40) correspond to [M − Zn]^+^ and [M − L − Pt]^+^ fragments for **C2** and **C3**. The obtained signals of molecular ions detected at 1145.94 (±0.24) correspond to [M + NH_4_^+^]^+^ adducts for **C4** and **C5**. The signals of fragment ions were also detected.

Conductivity measurements were performed using a 1 mM solution of the **C2**–**C5** complexes at 25.3 °C in solvents dimethylformamide and acetonitrile, as well. These solvents are preferred for conductivity measurements due to their high dielectric constant. Conductivity values for **C2** and **C3** in DMF are 131.4 and 155.9 Ω^−1^cm^2^ mol^−1^, respectively, while in acetonitrile, 251.2 and 230.1. For the **C4** and **C5** complexes, the obtained results in DMF are 157.2 and 148.0 Ω^−1^cm^2^ mol^−1^, and in acetonitrile, are 273.2 and 225.3 Ω^−1^cm^2^ mol^−1^, respectively. Based on the literature data, the obtained results are in agreement with the 2:1 electrolyte type and support the cationic nature of the complexes [17].

The results from characterization confirm the formation of heteronuclear complexes with bridging ligands pyrazine and 4,4′-bipyridyl.

### 2.2. CT-DNA-Binding Properties

DNA is a crucial target for anticancer drugs. Precise knowledge of drug–DNA binding is essential for drug development. Examined complexes interactions with CT-DNA were studied using UV–Vis spectroscopy, fluorescent emission titration, and docking measurements.

#### 2.2.1. Electronic Absorption Method

Absorption spectroscopic measurements enable the determination of whether complexes interact with the CT-DNA molecule. To explore this interaction, absorption titration was conducted using a fixed complex concentration of 8 µM and an increasing CT-DNA concentration up to a ratio of 5. As the CT-DNA concentration increases, the absorption intensity could either increase or decrease (hypochromic or hyperchromic shift), along with a minor shift in the absorption maximum wavelength [18,19,20,21,22]. The corresponding wavelength was monitored for absorbance during the increasing CT-DNA concentrations, as shown in Figure 2 and Appendix A, demonstrating a hyperchromic effect and a slight redshift. The values for the constant *K*_b_ can be found in Table 1. It was found that the absorbance properties of complexes **C4** and **C5** coincide with the range studied for the interaction with the CT-DNA molecule. Therefore, the measured absorbance values of the CT-DNA solution were corrected and attributed to the absorbance of the respective complexes. The corrected values are listed in Table 1.

Upon examination, all complexes displayed suitable *K*_b_ constant values. The constants value follows the order of growing **C3** < **C1** < **C2** < **C5** < **C4**. Notably, complex **C4** exhibited a markedly higher binding constant value than the other complexes assessed. Furthermore, complexes featuring transplatin exhibited a lower *K*_b_ constant value compared to cisplatin analogs. Conversely, complexes with bipyridine ligand components demonstrated higher *K*_b_ constant values when compared to pyrazine analogs.

#### 2.2.2. Fluorescence Spectroscopic Methods

Fluorescent spectroscopy allows the determination of the intercalation potential of compounds. Intercalation involves planar aromatic molecules binding between the base pairs of DNA chains, causing them to elongate. Ethidium bromide (EB) is commonly used as a DNA intercalator and produces high levels of fluorescent emissions at 612 nm when intercalating between DNA strands [25]. However, the addition of compounds that can intercalate between DNA strands and suppress the effects of EB will reduce fluorescent emissions [26,27,28]. If a complex is able to replace EB, fluorescence reduction will occur, indicating the complex’s ability to bind intercalatively. Figure 3 and Appendix A show the results of fluorescence titrations in the presence of intercalator EB, where the emission intensity of a constant CT-DNA/EB solution concentration of 5 µM was monitored with increasing complex concentration up to a ratio of 5.

To determine whether the complex could displace EB from the CT-DNA-EB complex, the interactions of the complex with CT-DNA were examined in the presence of EB. The CT-DNA-EB solution was formed by combining EB and CT-DNA solutions with a concentration of 5 μM at a pH of 7.4. The impact of the complex binding to the CT-DNA was measured by analyzing the fluorescence emission spectrum after the addition of the examined complexes solution in increasing concentration, Figure 3. Stern–Volmer constants, *K*_sv_, were obtained using Equation (S2) and are presented in Table 1

The binding constants of all the examined complexes were found to be in the order of 10^4^, indicating a solid intercalative ability. The values of the constants follow a specific order, with **C1** < **C3** < **C2**, **C5** < **C4.** Complex **C4** exhibited the highest binding constant value. It was observed that complexes containing transplatin had lower binding constant values compared to analog cisplatin. On the other hand, complexes containing bipyridine ligand parts displayed higher constant values than pyrazine complexes. It is worth noting that compared to the values obtained from UV–Vis, they were approximately two times higher.

### 2.3. Protein Binding Studies

Within the bloodstream, albumin is the most prevalent protein and plays a crucial role in transporting vital medications to their intended sites. Examining the interactions between the examined complexes and serum albumin provides a better understanding of the mechanism of action of anticancer drugs and highlights the significance of studying the interactions between serum albumin and different compounds to develop more effective cancer treatments.

#### Fluorescence Spectroscopy of HSA

Human serum albumin (HSA) produces a fluorescence signal when excited at 295 nm due to the tryptophan residues [29,30,31,32]. This characteristic makes fluorescence spectroscopy an effective tool for studying complex interactions involving HSA. If the examined complex interacts with HSA, an increasing concentration of the complex will lead to a decrease in fluorescence, as shown in Figure 4 and Appendix A. The decrease in fluorescence is indicative of changes in the tertiary protein structure, suggesting the binding of the examined complex to the HSA molecule [29,30,31,32]. The Stern–Volmer equation (Equation (S2)) was used to determine the Stern–Volmer constant, *K*_sv_, for the HSA interactions. The values of *K*_sv_, which are obtained from the linear dependence of I_0_/I to [Q], are provided in Table 1.

Based on the obtained *K*_sv_ values, it is evident that all examined complexes displayed a suitable binding affinity for the HSA molecule. The constants value increases following order **C1** < **C3** < **C2** < **C5** < **C4**. Complex **C4** exhibited the highest *K*_sv_ value, with a range of 10^5^ M^−1^, indicating a remarkable binding affinity for the HSA molecule. Conversely, the lowest *K*_sv_ value belonged to complex **C1**. It was observed that complexes containing transplatin had lower constant values compared to their cisplatin analogs. Moreover, complexes with bipyridine ligand components displayed higher constant values than pyrazine complexes.

When comparing the constants of the heteronuclear complexes **C2**–**C5** with those of the complexes from which the newly synthesized complexes were derived (transplatin, cisplatin, and the **C1** complex), it is observed that complexes **C2**, **C4**, and **C5** (Table 1) exhibit significantly enhanced constant values compared to the initial complexes. Nevertheless, complex **C2** maintains values akin to the starting complexes. These findings suggest that the construction of heteronuclear complexes contributes to improved reactivity, particularly in the instances of complexes **C2**, **C4**, and **C5**, while complex **C2** demonstrates reactivity similar to that of the starting complexes [23,24].

A comparison of results obtained from structurally closely related heteronuclear Pt(II)-Zn(II) complexes [11] reveals that the investigated complexes in Table 1 display analogous reactivity during their interaction with CT-DNA molecules. Moreover, both the heteronuclear Pt(II)-Pd(II) complexes and homonuclear Pt(II) and Pd(II) complexes exhibit comparable constant values when interacting with CT-DNA and proteins [28], mirroring the properties of the studied complex. This observation strongly indicates that dinuclear complexes generally possess binding constants on the order of 10^4^, signifying robust binding with both DNA molecules and albumin.

### 2.4. Analysis of Molecular Docking

#### 2.4.1. DNA Interaction

The molecular docking method was undertaken to predict the binding modes of the complexes to DNA. Calculated data of molecular docking are represented in Table 2. In the case of complex **C1,** the value of Δ*G* is −6.21 kcal·mol^−1^ with *K*_i_ of 28.18 µM. Further, **C2**–**C5** complexes yielded higher negative energy values (**C2** = Δ*G* = −9.76 kcal·mol^−1^; **C3** = Δ*G* = −10.01 kcal·mol^−1^; **C4** = Δ*G* = −10.80 kcal·mol^−1^; and **C5** = Δ*G* = −11.22 kcal·mol^−1^), with corresponding inhibition constants (**C2** = *K*_i_ = 28.12 µM; **C3** = *K*_i_ = 70.24 nM; **C4** = *K*_i_ = 12.10 nM; and **C5** = *K*_i_ = 5.93 nM).

In comparison with our previous results where one chlore was invoked into the system (Table 2), examined complexes **C2**–**C5** (with *tert*-butyl group inside) have higher negative energy values (from −9.76 to −11.20 kcal mol^−1^) than in the case of **C1a**–**C4a** (from −4.92 to −5.84 kcal mol^−1^) complexes. Inhibition constants for **C2**–**C5** are lower (70.24, 45.85, 12.10 and 5.93 nM) than for compounds **C1a**–**C4a** (52.15, 117.06, 248.49 and 230.84 µM). It could be said that the complex **C1** has the closest values compared to the previously examined (6.21 kcal·mol^−1^ with *K*_i_ of 28.18 µM).

Moreover, in comparison with other synthetized complexes, in the past (**C1b**–**C4b**) [6] (without invoking one Cl or *tert*-butyl group inside), interaction is better in the case of **C1**–**C5** complexes with DNA (Table 2). Compounds **C3b** and **C4b** show suitable binding energies −8.18 and −8.07 kcal mol^−1^ [11], but better values are obtained with **C2**–**C5** (−9.76, −10.01, −10.80, and −11.22 kcal mol^−1^). The trend is also better when it comes to constants. *K*_i_ values are in the range of µM in the case of **C1b**–**C4b** complexes (1.01, 1.22, 18.05, and 16.30 µM) and in the range of nM in the case of **C2**–**C5** (70.24, 45.85, 12.10, and 5.93 nM).

Thus, the represented data (Table 2) predict stronger binding to DNA in the case of **C2**, **C3**, **C4,** and **C5** complex systems. Obviously, with the involvement of the *tert*-butyl group in the system, interactions of a stronger intensity occur (higher negative Δ*G* energy and lower *K*_i_ values (see Table 2)). The best docking poses are presented in Figure 5. For accuracy purposes, we have let *bis*(1,10-phenanthroline)platinum(II) molecule (Appendix A) dock to 7DJW.

As far as interaction is concerned, in the case of the **C1** complex, the primary contribution comes from van der Waals forces. Further, in the case of **C2**, contributions are provided by conventional hydrogen bonds and by van der Waals forces. They are followed by **C3**–**C5** systems (Figure 6).

#### 2.4.2. HSA Interactions

Computational docking was also used to find the favored binding site and affinity between HSA and **C1**–**C2** complexes. For accuracy purposes, the validation method was employed with thyroxine (THYR) molecule (originally complexed in 1HK1 serum albumin, described elsewhere [18]). Table 3 displays the best docking scores for tested complexes in comparison to THYR. The highest binding affinities were observed for **C2** (Δ*G* = −43.30 kJ·mol^−1^) and **C3** (Δ*G* = −43.05 kJ·mol^−1^), while **C1** system exhibited slightly lower affinity (Δ*G* = −37.02 kJ·mol^−1^). On the other hand, complexes **C4** (Δ*G* = −29.95 kJ·mol^−1^) and **C5** (Δ*G* = −29.58 kJ·mol^−1^) demonstrated the weakest interaction. Therefore, the site I have chosen binds tighter in the case of **C1**, **C2,** and **C3** complexes.

However, in the case of **C4** and **C5** compounds, bonding can be expected on both sides (Table 3). The best docking poses of HSA with **C1**–**C5** complexes are shown in Figure 7.

When it comes to interaction in the **C1** complex, contributions primarily come from van der Waals forces, π-σ, alkyl, and π-alkyl bonds. In **C2** and **C3** compounds, the following interactions dominate: conventional hydrogen, π-σ, alkyl, and π-alkyl. Further, **C4** and **C5** systems show conventional hydrogen bonds, π-σ, and π-donor contributions.

### 2.5. Results of Antitumor Assessment

#### 2.5.1. MTT Assay: Evaluation of In Vitro Cytotoxicity of the Heteronuclear Platinum(II)/Zinc(II) Complexes

The in vitro cytotoxicity of newly synthesized mono- and heteronuclear platinum(II)/and zinc(II) complexes, featuring a 4,4′,4″-tri-*tert*-butyl-2,2′:6′,2″-terpyridine ligand (designated as **C1**–**C5**), was assessed using MTT assay procedures. This evaluation targeted various cell lines: mouse colorectal carcinoma (CT26), human colorectal carcinoma (HCT116 and SW480), and non-cancerous mouse mesenchymal stem cells (mMSC). These cell lines were exposed to synthesized complexes for 48 h, with concentrations ranging from 1.17 to 150 μM. Results demonstrated that the mono- and heteronuclear platinum(II) and zinc(II) complexes with the specified ligand (**C1**–**C5**) exhibited significant cytotoxic effects on both human (HCT116 and SW480) and mouse (CT26) colorectal carcinoma cells (Figure 8). Furthermore, the complexes **C1**–**C5** displayed dose-dependent cytotoxic activity against colorectal cancer cells while exhibiting relatively low cytotoxicity toward non-cancerous mMSC (Figure 8D). Comparatively, similar cytotoxic outcomes were observed with zinc(II)–terpyridine complexes, which reduced cell viability in various cancer cell lines including lung adenocarcinoma (A549), hepatocellular carcinoma (Bel-7402), breast adenocarcinoma (MCF-7), and esophageal squamous carcinoma (Eca-109), as referenced in the study by Li et al. [33]. Likewise, platinum(II)–terpyridine complexes exhibited significant antiproliferative effects across various cell lines, such as human squamous cell carcinoma (A431), human cervical cancer (HeLa), human breast cancer (MCF-7), human non-small-cell lung carcinoma (A549), and the cisplatin-resistant A549 subline (A549/DDP) [34].

In the context of developing low-toxicity drugs, it is imperative to evaluate the antiproliferative activity against normal cells. The in vitro toxicity of the tested complexes toward normal mouse mesenchymal stem cells (mMSC) was examined. All complexes were found to decrease the viability of non-cancer cells (mMSC) in a dose-dependent manner (Figure 8D). Notably, the **C1** complex demonstrated a lower cytotoxic effect on mMSC at concentrations ranging from 1.17 to 18.75 μM, compared to its effect on cancer cells (HCT116, SW480, and CT26). This reduced cytotoxicity of complex **C1** toward mMSC, coupled with its significant impact on the viability of tumor cells (HCT116, SW480, and CT26), suggests a potential selective cytotoxicity of this novel complex toward cancerous cell lines. These findings indicate the possibility of enhanced in vivo tolerance for this complex.

In the investigation into the cytotoxic efficacy of the **C1**–**C5** complexes, a thorough quantification of their half-maximal inhibitory concentration (IC_50_) values was performed. This evaluation targeted the murine colorectal carcinoma cell line CT26 and human colorectal carcinoma cell lines HCT116 and SW480. The obtained IC_50_ values, which indicate the concentration required by each complex to inhibit 50% of the cell viability, are meticulously cataloged in Table 4. Upon analysis of these IC_50_ values, it was observed that the **C1** complex exhibited enhanced cytotoxic effects against the murine CT26 and human HTC116 cells in comparison to the other complexes in the series. In contrast, cisplatin (CDDP), a standard chemotherapeutic agent, demonstrated more potent cytotoxic effects toward these tumor cells compared to the **C1**–**C5** complexes. However, it is noteworthy that cisplatin also exhibited increased cytotoxicity toward mouse mesenchymal stem cells, as detailed in Table 4.

Additionally, the therapeutic selectivity of these complexes was evaluated by calculating the selectivity index, presented in Table 5. This index is a vital metric for determining the relative cytotoxicity of a compound on cancer cells versus non-cancerous cells. Calculated as the ratio of the IC_50_ value in mouse mesenchymal stem cells to that in tumor cells, a higher selectivity index is indicative of greater specificity and reduced toxicity toward normal cells, a key consideration in cancer therapy. A significant outcome of this study was the markedly high selectivity index observed for the **C1** complex. This suggests that the **C1** complex possesses a notably higher cytotoxic effect on the tested cancer cell lines (CT26, HCT116, and SW480) relative to non-cancerous mMSC. This pronounced selectivity underscores the potential of the **C1** complex as a candidate for targeted cancer therapy, implying a reduced risk of damage to healthy cells. Based on these observations, the **C1** complex was chosen for further in-depth analysis.

#### 2.5.2. Assessment of Apoptotic Cell Death Induced by **C1** Complex

Apoptosis, commonly referred to as programmed cell death, is a highly regulated cellular process that occurs in a controlled manner within individual cells, ensuring no damage to neighboring cells. This precision makes apoptosis induction a vital strategy in carcinoma treatment, as it targets and systematically eliminates cancer cells while preserving healthy tissue. The induction of apoptosis disrupts the abnormal proliferation characteristic of cancer cells, thereby inhibiting tumor growth and progression. This method’s effectiveness lies in its ability to selectively target cancer cells, reducing the likelihood of adverse effects commonly associated with other cancer therapies [35].

The capacity of the **C1** complex to induce apoptotic death in cancer cells was assessed using flow cytometric analysis. This analysis involved treating cells with the **C1** complex and subsequently staining them with Annexin V FITC and Propidium Iodide. The results, illustrated in Figure 9, revealed that a significant proportion of CT26 cells, after 24 h treatment with the **C1** complex, entered both early and late stages of apoptosis. Moreover, a noteworthy observation was that a substantial percentage of the CT26 cells also underwent necrosis in comparison to untreated cells, as depicted in the same Figure 9. These findings highlight the dual action of the **C1** complex in promoting cell death through both apoptotic and necrotic pathways, suggesting its potential as a potent anti-carcinoma agent. In a similar vein, in a related context, NCI-H460 cancer cells, which are a line of human large-cell lung carcinoma cells, were subjected to treatment with binuclear platinum(II) complexes. These complexes feature 4′-substituted-2,2′:6′,2″-terpyridine ligands and were administered for a duration of 24 h. Following this treatment, there was a significant increase in the proportion of apoptotic cells, correlating strongly with exposure to these specific ligands [36].

Apoptosis, a critical form of programmed cell death, plays a pivotal role in maintaining cellular homeostasis and is essential for the development and health of multicellular organisms. Apoptosis is primarily induced via two pathways: the extrinsic apoptotic pathway, which involves death receptors leading to the activation of caspase-3, and the intrinsic or mitochondrial apoptotic pathway [37]. The latter is triggered by an imbalance between pro-apoptotic and antiapoptotic proteins, such as Bax and Bcl-2. This imbalance alters the permeability of the mitochondrial membrane, resulting in the release of cytochrome c from mitochondria and the subsequent activation of caspase-3, leading to apoptosis [38]. Bax (Bcl-2-associated X protein), a pro-apoptotic member of the Bcl-2 protein family, promotes cell death by facilitating the release of cytochrome c from mitochondria [39]. In contrast, Bcl-2 (B-cell lymphoma 2) functions as an anti-apoptotic protein, inhibiting the activity of pro-apoptotic agents and thus preventing apoptosis [40]. Caspase-3, an essential executioner caspase in the apoptosis pathway, is activated in apoptotic cells by both extrinsic (death ligand) and intrinsic (mitochondrial) pathways and is responsible for cleaving various key cellular proteins, leading to the morphological and biochemical changes characteristic of apoptosis [41].

Our study’s findings indicate that treatment of cells with IC_50_ concentrations of the **C1** complex increased the percentage of Bax-positive CT26 cells while decreasing the percentage of Bcl-2-positive CT26 cells (Figure 10A,B). Additionally, the **C1** complex was observed to increase the proportion of caspase-3-positive CT26 cells (Figure 10C). The observed increase in caspase-3 expression coupled with a decrease in Bcl-2 expression suggests potential mechanisms of action for the **C1** complex via the induction of apoptosis. Furthermore, the significant increase in the Bax molecule, a pro-apoptotic agent, following treatment with the **C1** complex underscores its role in promoting apoptosis in CT26 cells. Similarly, studies have demonstrated that Zn(II) terpyridine-based nitrate complexes play a significant role in the process of apoptosis [42]. These complexes, akin to the **C1** complex, appear to modulate key factors in apoptotic pathways, contributing to the induction of cell death in cancerous cells.

#### 2.5.3. Effects of Compound C1 on Cell Cycle and Proliferation in CT26 Carcinoma Cells

Ki67, a well-established proliferation-associated protein, functions as a DNA-binding marker expressed predominantly in actively proliferating cells but not in cells in a quiescent state. It is recognized as a reliable indicator of cell proliferation due to its presence during all active phases of the cell cycle (G1, S, G2, and M phases) but is absent in the resting phase (G0) [43]. Ki67 is often used in clinical and research settings to assess the growth fraction of a cell population, thereby providing insights into the rate of cell proliferation. Its expression level is directly correlated with cell proliferation, making it a valuable tool for evaluating the proliferative status of cancer cells and the effectiveness of anticancer treatments [44]. In the context of evaluating the antiproliferative effects of the tested **C1** complex, the detection of Ki67 expression levels in treated CT26 cells was undertaken. The study observed a significantly lower expression of the Ki67 molecule in CT26 cells following treatment with the **C1** complex compared to untreated control cells, as illustrated in Figure 11A. This reduction in Ki67 expression suggests a marked decrease in cell proliferation.

The study of cell cycle arrest in CT26 carcinoma cells treated with the potent compound **C1** was conducted to elucidate the potential cellular mechanisms underlying its anticancer effect. For this purpose, CT26 cells were subjected to 24 h treatment with the **C1** complex. Our findings reveal that the newly synthesized **C1** complex effectively induced a G0/G1 phase arrest in CT26 colorectal carcinoma cells (Figure 11B). This arrest is likely attributable to molecular alterations within the cancer cells post-treatment. In addition to the G0/G1 phase arrest, a notable decrease in the synthesis (S) phase was observed following the treatment with the **C1** compound (Figure 11B). This reduction in the S phase, where DNA replication occurs, further indicates the efficacy of the **C1** complex in hindering cell cycle progression. This interruption of the cell cycle progression suggests that the **C1** complex disrupts the normal process of cell division, an essential mechanism for the proliferation of cancer cells. The induction of cell cycle arrest by the **C1** compound potentially curtails the growth and dissemination of carcinoma cells, underscoring its efficacy as a promising anticancer agent.

Cyclins play a pivotal role in the regulation of the cell cycle. They are a family of proteins that control the progression of cells through the cell cycle by activating cyclin-dependent kinases (CDKs). The activity of cyclins varies throughout the different phases of the cell cycle, and their regulated expression ensures the proper progression and timing of the cycle [45]. Among the various types of cyclins, Cyclin D and Cyclin E are particularly crucial during the G1/S transition [46]. Cyclin D partners with CDK4 and CDK6, playing a key role in the transition from the G1 phase to the S phase of the cell cycle by promoting the cellular environment necessary for DNA replication. It is often regarded as a sensor of extracellular mitogenic signals and is integral in the decision of a cell to divide [47]. On the other hand, Cyclin E associates with CDK2 and is instrumental in the preparation for DNA synthesis, helping to drive the cell into the S phase. The regulated expression of Cyclin D and Cyclin E is essential for maintaining normal cell cycle progression, and their dysregulation is frequently observed in various cancers [48].

To evaluate the impact of the **C1** complex on Cyclin D and Cyclin E, CT26 cells were treated with an IC_50_ concentration of the **C1** compound for 24 h. The results, illustrated in Figure 12A, demonstrate that the **C1** complex significantly reduced the percentage of Cyclin D-positive cells, indicating a decrease in the population of cells transitioning from the G1 to the S phase. However, the expression of Cyclin E-positive cells was not notably altered by the treatment (Figure 12B). This selective downregulation of Cyclin D suggests that the **C1** complex specifically impedes the cell cycle progression at the G1 phase, potentially leading to cell cycle arrest and inhibiting further proliferation of the CT26 cells. In a similar context, it has been found that ruthenium(II)–platinum(II) bis(terpyridyl) anticancer complexes also alter the activities of Cyclin D and Cyclin E. These complexes have been shown to block the entry of cells into the S phase [49].

There is a critical interplay between Cyclin D, p21, and phospho-AKT (p-AKT) in cell cycle regulation. p21, a cyclin-dependent kinase (CDK) inhibitor, plays a pivotal role in cell cycle control by inhibiting the activity of CDK–Cyclin complexes, including those involving Cyclin D. p21 can be induced in response to various stimuli, including DNA damage and other stress signals, acting as a regulator to halt cell cycle progression, particularly at the G1 checkpoint. This inhibition is a crucial aspect of the cell’s response to damage, preventing the replication of damaged DNA [50]. On the other hand, p-AKT, a phosphorylated form of protein kinase B (AKT), is a significant oncogenic player involved in the control of cell proliferation and survival. AKT phosphorylation leads to the activation of downstream pathways that promote cell growth and survival, making it a critical target in cancer therapy. The dysregulation of AKT signaling is frequently observed in various cancers, contributing to tumorigenesis and the development of resistance to therapies [50].

Upon treatment with the **C1** complex, CT26 cells exhibited changes in the expression of these crucial cell cycle regulators. Notably, there was an increase in the proportion of cells expressing p21, a CDK inhibitor (Figure 12C). This upregulation of p21 is likely to contribute to the suppression of CDK activity, thereby reinforcing cell cycle arrest. Additionally, the C1 complex treatment was observed to cause a marked decrease in the levels of phosphorylated AKT (p-AKT) (Figure 12D). The reduction in p-AKT, a crucial component in oncogenic signaling pathways that facilitate cell survival and proliferation, suggests a potential antitumor effect of the **C1** complex. These changes in both p21 and p-AKT expression indicate the **C1** complex’s capability to disrupt critical processes in cell cycle regulation and survival signaling, thereby exhibiting its potential as an effective anticancer agent.

Overall, it could be concluded that the substituents on tridentate terpyridine ligands, such as *tert*-butyl groups, significantly influence the antiproliferative effect, DNA binding mode of mononuclear and heteronuclear platinum(II)/zinc(II)complexes. The steric hindrance caused by attaching bulky *tert*-butyl groups to terpyridine increases the size of heteronuclear complexes, reducing their flexibility and reactivity. This could be the main reason why the mononuclear **C1** complex, in comparison with the heteronuclear **C2**–**C5**, exhibits the highest selectivity index, making it a promising targeted anticancer agent.

## 3. Materials and Methods

### 3.1. Materials

All reagents used in this study were analytical grade or higher purity, obtained from Sigma-Aldrich (St. Louis, MO, USA), Acros Organics (Thermo Fisher Scientific, Geel, Belgium), and Merck (Darmstadt, Germany).

### 3.2. Instrumentations

Elemental analyses were performed on a Carlo Erba Elemental Analyzer 1106 and Atomic Absorption Spectrometers novAA 400P (Analytik Jena, Jena, Germany). The UV/Vis absorption spectra were recorded on Uvikon XS (Secomam, Alès, France) and Perkin Elmer UV/Vis Lambda 35 (Perkin Elmer Inc., Shelton, CT, USA), double beam UV/Vis spectrophotometers packing pre-aligned tungsten and deuterium lamps, wavelength range of 190–1100 nm and a variable bandwidth range of 0.5 to 4 nm) equipped with a 1 cm path length cell. IR data were obtained using a PerkinElmer^®^ Spectrum One FT-IR spectrometer (Perkin Elmer Inc., Shelton, CT, USA). The ^1^H spectra were acquired on a Varian Gemini-2000 spectrometer (200 MHz) (Varian, Inc. Palo Alto, CA, USA). ESI (electrospray ionization) mass spectra were recorded on an LTQ Orbitrap XL (Thermo Fisher Scientific, Waltham, MA, USA) using direct injection of the complex solution in acetonitrile. The pH measurements were recorded on a Jenway 4330 pH meter (Thermo Fisher Scientific, Waltham, MA, USA) with a combined Jenway glass microelectrode that had been calibrated with standard buffer solutions of pH 4.0, 7.0, and 10.0 (Merck). The KCl solution in the reference electrode was replaced with a 3 M NaCl electrolyte to prevent precipitation of KClO_4_ during use [51,52,53,54].

### 3.3. Synthesis of Complexes

The complex [ZnCl_2_(terpy*^t^*^Bu^)] was synthetized according to the literature procedure [55]. The complex was recrystallized from dimethylformamide (DMF). The powder was washed with ether and then dried in a vacuum [55].

#### 3.3.1. Preparation of *Cis*- or *Trans*-[PtCl(NH_3_)_2_(H_2_O)](ClO_4_)

To a stirred aqueous solution of *cis*- or *trans*-diamminedichloridoplatinum(II) 0.0279 g (0.093 mmol, 10 mL), aqueous solution of AgClO_4_ 0.019 g (0.093 mmol, 10 mL) was added drop-wise. The mixture was stirred overnight in the dark at 40 °C, and the AgCl precipitate was then filtered off. The resulting pale yellow solution of *cis*- or *trans*-[PtCl(NH_3_)_2_(H_2_O)](ClO_4_) was used as the starting material for the preparation of the heteronuclear Pt-L-Zn complexes.

#### 3.3.2. Preparation of [ZnCl(terpy^tBu^)(H_2_O)]ClO_4_

To a stirred aqueous solution of [ZnCl_2_(terpy*^t^*^Bu^)] 0.05 g (0.093 mmol, 10 mL), an aqueous solution of AgClO_4_ 0.019 g (0.093 mmol, 10 mL) was added drop-wise. The mixture was stirred at 40 °C overnight in the dark, and the AgCl precipitate was then filtered off. The solution of [ZnCl(terpy*^t^*^Bu^)(H_2_O)](CIO_4_) was used as the starting material for the heteronuclear Pt(II)-L-Zn(II) complexes preparation.

#### 3.3.3. Preparation of the Heteronuclear Platinum(II)/Zinc(II) Complexes

To a stirred aqueous solution of *cis-* or *trans*-[PtCl(NH_3_)_2_(H_2_O)](ClO_4_) 0.0279 g (0.093 mmol, 20 mL), an aqueous solution of the bridging ligand (pyrazine or 4,4′-bipyridyl) (0.093 mmol, 20 mL) was added drop-wise at room temperature. After 1 h, the solution of [ZnCl(terpy*^t^*^Bu^)(H_2_O)](CIO_4_) complex 0.05 g (0.093 mmol, 20 mL) was added drop-wise. The pH was adjusted between 4 and 5 using 0.1 M HClO_4_, and the reaction mixture was left to stir overnight. The resulting colored solution was filtered and kept aside for slow evaporation at room temperature. The color of the complex powders was yellow-white when the bridging ligand was pyrazine, while white-orange when 4,4′-bipyridyl was used. The obtained complexes were recrystallized from dimethylformamide (DMF). The powders were washed with small amounts of ether and then dried under vacuum.

**[ZnCl_2_(terpy*^t^*^Bu^)] (C1)**: Yield 0.433 g (81%) Anal. Calcd. for C_27_H_35_Cl_2_N_3_Zn: N 7.81; C 60.29; H 6.56; Found: N 7.77; C 60.05; H 6.53%. **^1^**H NMR (DMSO-d6, 295 K): δ 8.77 (s, 2H, terpyridine), 8.68 (t, 2H, terpyridine), 8.45 (s, 2H, terpyridine), 7.55 (t, 2H, terpyridine), 1.56 (s, 27H, *t*Bu) ppm. FT-IR (KBr): 2962 ν(C-H), 1589, 1548, 1477, and 1374 ν(C=C, C=N, aromatic ring), 1267, 1015ν(C-N), 865 ν(C=C), 614 ν(Zn-N) cm^−^^1^. UV–Vis (H_2_O, **λ**_max_, nm): 276, 280. ESI-MS: *m*/*z* 577.99 ([M + K]^+^) calcd. 577.13.

**[{*cis*-PtCl(NH_3_)_2_(*μ*-pyrazine)ZnCl(terpy*^t^*^Bu^)}](ClO_4_)_2_ (C2)**: Yield 0.059 g (60.82%) Anal. Calcd. for C_31_H_45_Cl_4_N_7_O_8_PtZn: N 9.37; C 35.6; H 4.34; Found: N 9.33; C 35.46; H 4.33. ^1^H NMR (DMSO-*d_6_*, 25 °C): 9.25–9.22 (m, 2H, terpyridine), δ 8.99–8.95 (m, 4H, pyrazine), 8.92 (m, 2H, terpyridine), 7.33 (s, 2H, terpyridine), 7.08 (s, 1H, terpyridine), 6.84 (s, 1H, terpyridine), 2.86 (s, 3H, NH_3_), 1.48 (s, 27H, *t*Bu) ppm. FT-IR (KBr): 3436 and 3293 ν(N-H, NH_3_), 3116 ν(C-H), 1619, 1435, and 1342 ν(C=C, C=N, aromatic ring), 1086 ν(C-N), 820 ν(C=C), 626 ν(Zn-N) cm^−^^1^. UV–Vis (H_2_O, **λ**_max_, in nm): 274, 284, 306, 316, and 328. Molar conductivity (Λ_M_, DMF): 131.4 Ω^−^^1^cm^2^mol^−^^1^, (Λ_M_, Acetonitrile): 251.2 Ω^−^^1^cm^2^mol^−^^1^. ESI-MS: *m*/*z* 344.02 ([M − Zn]^+^) calcd. 344.16; 488.04 ([M – L − Pt]^+^) calcd. 488.21.

**[{*trans*-PtCl(NH_3_)_2_(*μ*-pyrazine)ZnCl(terpy*^t^*^Bu^)}](ClO_4_)_2_ (C3)**: Yield 0.064 g (75.1%) Anal. Calcd. for C_31_H_45_Cl_4_N_7_O_8_PtZn: N 9.37; C 35.60; H 4.34; Found: N 9.35; C 35.51; H 4.35. ^1^H NMR (DMSO-*d_6_*, 25 °C): 9.05–9.00 (m, 2H, terpyridine), δ 8.95–8.74 (m, 2H, pyrazine), 8.10 (m, 2H, terpyridine), 7.33 (s, 2H, terpyridine), 7.07 (s, 1H, terpyridine), 6.82 (s, 1H, terpyridine), 3.18 (s, 3H, NH_3_), 1.23 (s, 27H, *t*Bu) ppm. FT-IR(KBr): 3436 and 3307 ν(N-H, NH_3_), 2923 ν(C-H), 1617 and 1423 ν(C=C, C=N, aromatic ring), 1086 ν(C-N), 820 ν(C=C), 623 ν(Zn-N) cm^−^^1^. UV–Vis (H_2_O, λ_max_, in nm): 274, 284, 318, and 330. Molar conductivity (Λ_M_, DMF): 155.9 Ω^−^^1^cm^2^mol^−^^1^; (Λ_M_, Acetonitrile): 230.1 Ω^−^^1^cm^2^mol^−^^1^. ESI-MS: *m*/*z* 344.22 ([M − Zn]^+^) calcd. 344.16; 488.44 ([M − L − Pt]^+^) calcd. 488.21.

**[{*cis*-PtCl(NH_3_)_2_(*μ*-4,4′-bipyridyl)ZnCl(terpy*^t^*^Bu^)}](CIO_4_)_2_ (C4)**: Yield 0.07 g (66.9%) Anal. Calcd. for C_37_H49Cl4N7O8PtZn: N 8.74; C 39.60; H 4.40 Found: N 8.71; C 39.50; H 4.39%. **^1^**H NMR (DMSO-*d*_6_, 25 °C): δ 8.98–8.84 (m, 4H, 4-pyridine, 2H terpyridine), 8.36 (m, 2H, terpyridine, 4H, 4-pyridine), 7.99 (s, 2H, terpyridine), 7.95 (s, 1H, terpyridine), 7.85 (s, 1H, terpyridine), 4.75 (s, 3H, NH_3_), 1.23 (s, 27H, *t*Bu) ppm. FT-IR (KBr): 3436 v(N-H, NH_3_), 2962 ν(C-H), 1607, 1553, 1548, 1477, and 1404 ν(C=C, C=N, aromatic ring), 1254 and 1015 ν(C-N), 866 ν(C=C), 615 ν(Zn-N) cm^−^^1^. UV–Vis (H_2_O, **λ**_max_, in nm): 282. Molar conductivity (Λ_M_, DMF): 157.2 Ω^−^^1^cm^2^mol^−^^1^, (Λ_M_, Acetonitrile): 237.2 Ω^−^^1^cm^2^mol^−^^1^. ESI-MS: *m*/*z* 1145.70 ([M + NH_4_^+^]) calcd. 1145.20.

**[{*cis*-PtCl(NH_3_)_2_(*μ*-4,4′-bipyridyl)ZnCl(terpy*^t^*^Bu^)}](CIO_4_)_2_ (C5)**: Yield 0.06 g (65.5%) Anal. Calcd. for C_37_H49Cl4N7O8PtZn: N 8.74; C 39.60; H 4.40 Found: N 8.70; C 39.44; H 4.38%. **^1^**H NMR (DMSO-*d_6_*, 25 °C): δ 8.99–8.84 (m, 4H, 4-pyridine, 2H terpyridine), 8.35–8.26 (m, 2H, terpyridine, 4H, 4-pyridine), 7.99 (s, 2H, terpyridine), 7.95 (s, 1H, terpyridine), 7.85 (s, 1H, terpyridine), 1.46 (s, 3H, NH_3_), 1.23 (s, 27H, *t*Bu) ppm. FT-IR (KBr): 3419 and 3309 v(N-H, NH_3_), 2932 ν(C-H), 1659, 1551, 1437, and 1391 ν(C=C, C=N, aromatic ring), 1089 ν(C-N), 821 ν(C=C), 627 ν(Zn-N) cm^−^^1^. UV–Vis (H_2_O, λ_max_, in nm): 278. Molar conductivity (Λ_M_, DMF): 148.0 Ω^−^^1^cm^2^mol^−^^1^, (Λ_M_, Acetonitrile): 225.3 Ω^−^^1^cm^2^mol^−^^1^. ESI-MS: *m*/*z* 1145.94 ([M + NH_4_^+^]) calcd. 1145.20.

### 3.4. Binding Interactions

#### 3.4.1. CT-DNA Binding Interactions

##### UV–Vis Spectroscopy Studies

The binding affinity of the complexes **C1**–**C5** to CT-DNA was determined using UV–Vis spectroscopy. Measurements were conducted at room temperature in a 0.01 M phosphate buffer saline, PBS (Thermo Fisher Scientific Inc., Waltham, MA, USA) with a pH of 7.4. PBS is a pH-adjusted phosphate buffer and saline solutions that, when diluted to a 1X working concentration, contains 137 mM NaCl, 2.7 mM KCl, 8 mM Na_2_HPO_4_, and 2 mM KH_2_PO_4_. A range of solutions containing both complex and CT-DNA were prepared, with increasing concentrations of CT-DNA (0–40 μM) added to a fixed concentration of the complex solution (8 μM). The intrinsic equilibrium binding constants, *K*_b_, were calculated using Equation (1).
[CT-DNA]/(ε_A_ − ε_f_) = [CT-DNA]/(ε_b_ − ε_f_) + 1/*K*_b_(ε_b_ − ε_f_)(1)
*K*_b_ is calculated from the slope and the y-intercept ratio [CT-DNA]/(ε_A_ − ε_f_) = f([CT-DNA]) (Figure 2), where [CT-DNA] is DNA concentration, ε_A_ = A_obsd_/[complex], ε_f_ is the extensional coefficient of the uncoordinated complex, and εb is the extinction coefficient of the coordinated complex [18,19,20,21,22].

##### Fluorescence Quenching Studies

The binding affinity of the complexes **C1**–**C5** to CT-DNA via intercalation was determined using fluorescence spectroscopy. All measurements were conducted at room temperature in 0.01 M PBS with a pH of 7.4. Fluorescence measurements were performed by adding increasing complex concentrations (0–50 μM) to a CT-DNA and EB solution. Before each measurement, the system was shaken at room temperature. The fluorescence intensity was measured with the wavelength of excitation at 527 nm and the wavelength of emission at 612 nm. The width of the excitation and emission slits (10 nm) and scan rate were maintained at a constant rate. The emission of the solution was recorded in the range of 550 to 750 nm. The Stern–Volmer constants, *K*_sv_, were determined by Equation (2).
I_0_/I = 1 + *K*_sv_[Q](2)
[Q] is the concentration of the complex, I_0_ and I represent emission intensities in the absence and presence of complex, and *K*_sv_ is the Stern–Volmer constant obtained from the slope of the plot of I_0_/I vs. [Q] [25,26,27,28].

#### 3.4.2. HSA Binding Interactions

The binding affinity of the complexes **C1**–**C5** to HSA has been determined through fluorescence quenching experiments. The measurements were conducted in a 0.01 M PBS with a pH of 7.4 at room temperature. The HSA concentration was kept constant at 2 μM while the concentration of complexes **C1**–**C5** was increased from 0 to 30 μM, resulting in a significant decrease in emissions at 352 nm. Before each measurement, the system was shaken at room temperature. The fluorescence intensity was measured with the wavelength of excitation at 295 nm and recorded in the 300 to 500 nm range. The width of the excitation and emission slits (10 nm) and scan rate were maintained at a constant rate. The Stern–Volmer constants, *K*_sv_, were determined by Equation (2).

### 3.5. Computational Methods

#### 3.5.1. DFT Calculations

The B3LYP functional [56,57] was used to optimize the geometries together with the LANL2DZ basis set in all examined complexes. The structures were visualized in the free version of the Discovery Studio Visualizer 3.5.0 (v17.2.0) (Accelrys Software Inc.,San Diego, CA, USA)[58]. These calculations were performed by the Gaussian 09 program package (Gaussian, Inc., Wallingford CT, USA, 2009). [59].

#### 3.5.2. Molecular Docking

The X-ray crystal structure of B-DNA (PDB ID: 1BNA) and HSA (PDB ID: 1HK1) was acquired from the Protein Data Bank (PDB) (RCSB PDB: Homepage). Docking processes were carried out using Autodock 4.2 [60] software equipped with the graphical user interface (GUI) Auto-DockTools (ADT 1.5.6rc3) [60]. Then, the polar hydrogen atoms were added, and ADT was used to remove crystal water. Geisteiger charges were added to each atom, and non-polar hydrogen atoms were merged into the DNA structure. The structures were then saved in PDBQT file format for further studies in ADT. For the visualization of the docking results, a free version of the Discovery Studio Visualizer 3.5.0 Accelrys Software Inc. [58], PyMOL program (v1.3r1) [61], and Chimera software (v 1.17.3) were used [62].

### 3.6. Antitumor Assessment

#### 3.6.1. Cell Culture

In our study, various cell lines were cultured, including mouse colorectal carcinoma (CT26), human colorectal carcinoma (HCT116 and SW480) obtained from ATCC, USA, and mouse mesenchymal stem cells (mMSC, Gibco), New York, NY, USA). These cell lines were consistently maintained in complete Dulbecco’s Modified Eagle Medium (DMEM) supplemented with 10% fetal bovine serum (FBS) (Sigma Aldrich, St. Louis, MO, USA), 100 IU/mL penicillin, and 100 μg/mL streptomycin antibiotics (Invitrogen, Carlsbad, CA, USA). The cultures were incubated at 37 °C in a humidified atmosphere of 5% carbon dioxide (CO_2_). Throughout the experiments, only cell suspensions exhibiting greater than 95% viability were utilized. The viability and count of the cells were determined using Trypan blue staining, ensuring the accuracy and reliability of the experimental outcomes.

#### 3.6.2. MTT Assay

The cytotoxic effects of five newly synthesized complexes, alongside cisplatin, were assessed through the MTT assay method, in line with previously established protocols [63]. During the exponential growth phase, cells were collected from culture flasks, counted, and then seeded in 96-well culture plates at a density of 5·10^3^ cells per well. Subsequent treatment involved exposing these cells to varying concentrations of the newly synthesized complexes and cisplatin (ranging from 1.17 to 150 μM) for a duration of 48 h, with fresh complete medium serving as the control. The results were quantified relative to the control (untreated cells), providing a comparative measure of cytotoxicity. Additionally, IC_50_ values, representing the concentration required to inhibit 50% of the cell viability, were calculated using Microsoft Office Excel 2010. Lastly, the data were presented as percentages of viable cells in accordance with the methodology outlined in the referenced study [64], facilitating a comprehensive understanding of the anticancer potential of these compounds.

#### 3.6.3. Assessment of Apoptosis

The evaluation of apoptosis in cells subjected to treatment was conducted employing the Annexin V and Propidium Iodide double staining assay in accordance with the methodology outlined in prior studies [65]. This approach enabled the determination of the percentage of apoptotic cells. In further experiments, cells were fixed and permeabilized using a permeabilization buffer provided by BD Bioscience, Heidelberg, Germany. Subsequently, they were incubated with specific antibodies targeting Bcl-2, Bax, and caspase-3, all sourced from Thermo Fisher Scientific Inc., Waltham, MA, USA, following the procedures described in previous research [65]. The analysis of these treated cells was carried out using a FACS Calibur flow cytometer (BD Biosciences, San Jose, CA, USA) and the resulting data were meticulously analyzed with FlowJo software (v10.8.2). This technique provided a comprehensive and detailed assessment of apoptotic responses in cells following exposure to various treatments, allowing for a thorough understanding of the apoptotic processes induced under experimental conditions.

#### 3.6.4. Cell Cycle Analysis

Our research progressed to explore the potential effects of the newly synthesized C1 complex on the cell cycle dynamics of CT26 tumor cells. The aim was to identify Ki67 expression as well as any alterations in the cell cycle progression of tumor cells treated with the **C1** complex compared to those that were not exposed to this compound. The experimental setup included a group of cells treated with concentrations equivalent to the IC_50_ values of the compounds and a control group of untreated cells. These cells were cultivated in 25 mL flasks. Post-treatment with the IC_50_ concentrations of the **C1** complex, the cells underwent a further incubation period of 24 h, followed by trypsinization, triple washing in PBS, and cell counting. Each sample tube was then treated with a specific antibody targeting Ki67 (eBioscience, San Diego, CA, USA)/1 μL of Vybrant^®^ DyeCycleTM Ruby stain (Thermo Fisher Scientific Inc., Waltham, MA, USAand subsequently analyzed using a FACS Calibur flow cytometer (BD Biosciences, San Jose, CA, USA). A minimum of 15,000 events per sample were examined. Data analysis was conducted using the FlowJo vX.0.7 software, allowing for a detailed assessment of the cell cycle phases and any impact induced by the **C1** complex.

#### 3.6.5. Evaluation of the Impact on Cell Cycle Regulators

In this part of the study, CT26 cancer cells, treated with the IC_50_ concentration of the **C1** complex or maintained in a culture medium alone as a control for 24 h, underwent a series of cellular analyses. Initially, the cells were fixed and permeabilized using a permeabilization buffer from BD Bioscience, Heidelberg, Germany. Following this, they were incubated with specific antibodies targeting key cell cycle regulators, namely Cyclin D, Cyclin E, p21, and phospho-AKT (p-AKT), all provided by Thermo Fisher Scientific. The analysis of these cells was performed using a FACS Calibur flow cytometer (BD Biosciences, San Jose, CA, USA), and the gathered data were subsequently processed and interpreted with FlowJo software (v10.8.2).

## 4. Conclusions

The novel [ZnCl_2_(terpy*^t^*^Bu^)] (**C1**), [{*cis*-PtCl(NH_3_)_2_(*μ*-pyrazine)ZnCl(terpy*^t^*^Bu^)}](ClO_4_)_2_ (**C2**), [{*trans*-PtCl(NH_3_)_2_(*μ*-pyrazine)ZnCl(terpy*^t^*^Bu^)}](ClO_4_)_2_ (**C3**), [{*cis*-PtCl(NH_3_)_2_(*μ*-4,4′-bipyridyl)ZnCl(terpy*^t^*^Bu^)}](CIO_4_)_2_ (**C4**), [{*trans*-PtCl(NH_3_)_2_(*μ*-4,4′-bipyridyl)ZnCl(terpy*^t^*^Bu^)}](CIO_4_)_2_ (**C5**) complexes were synthesized and characterized. All complexes displayed suitable *K*_b_ constant values. Constants value follows the order of growing **C3** < **C1** < **C2** < **C5** < **C4**. Notably, the **C4** complex exhibited a markedly higher binding constant for interaction with DNA than the other complexes assessed. Furthermore, complexes featuring transplatin exhibited a lower *K*_b_ constant value compared to cisplatin analogs. Based on the obtained *K*_sv_ values, it is evident that all examined complexes displayed a suitable binding affinity for the HSA molecule. The constants value increases following order **C1** < **C3** < **C2** < **C5** < **C4**. Complex **C4** exhibited the highest *K*_sv_ value, while the lowest *K*_sv_ value belonged to complex **C1**. The molecular docking studies confirm that the binding of complexes **C1** to DNA is due to van der Waals forces, while that of **C2**–**C5** is due to conventional hydrogen bonds and van der Waals forces.

The in vitro cytotoxicity evaluation of the newly synthesized **C1**–**C5** complexes, particularly the **C1** complex, demonstrated pronounced selectivity toward cancer cells over non-cancerous mMSC. The **C1** complex notably induced apoptosis in CT26 cells, underlined by increased Bax and caspase-3 expression and decreased Bcl-2 and Ki67 levels, suggesting a strong apoptotic and antiproliferative action. Furthermore, the **C1** complex effectively arrested the cell cycle in the G0/G1 phase and selectively down-regulated Cyclin D, while p21 and p-AKT expression changes pointed toward a disruption in cell cycle progression and survival signaling. These findings collectively highlight the **C1** complex’s potential as a targeted anticancer agent with a unique ability to modulate crucial cellular mechanisms in cancer cells. The steric hindrance caused by attaching bulky *tert*-butyl groups to terpyridine reduces the flexibility and antiproliferative ability of heteronuclear complexes.

## Data Availability

The data are available upon request.

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
