# Peer review of "Modes of Interactions with DNA/HSA Biomolecules and Comparative Cytotoxic Studies of Newly Synthesized Mononuclear Zinc(II) and Heteronuclear Platinum(II)/Zinc(II) Complexes toward Colorectal Cancer Cells"

_ijms, 2024, doi:10.3390/ijms25053027_

Round 1

Reviewer 1 Report

Comments and Suggestions for Authors

The cancer is still one of the main causes of death in humans. Therefore, it is so important to searching new diagnostic and therapeutic agents. One of the strategies is finding metal ion complexes. The discovery of cisplatin compounds sets a new direction for research of new anticancer therapies. Although cisplatin is currently used drug in cancer therapy it is severely limited by toxic side effects. For this reason, the search for new metal-dependent complex compounds that have anticancer properties and do not have side effects is an important issue nowadays. The authors of the manuscript: “Modes of interactions with DNA/BSA biomolecules and comparative cytotoxic studies of newly synthesized mononuclear zinc(II) and heteronuclear platinum(II)/zinc(II) complexes toward colorectal cancer cells” present research results on new ruthenium and zinc complexes, including heteronuclear complexes, which is a new approach. The article has a structure typical of research work. The introduction part is short, and quite well-written but could be extended. The work has great potential because it is an interdisciplinary approach. The authors present the history of compounds from synthesis through studies of interactions with macromolecules and biological evaluation. This is the biggest advantage of this work. However, the way of presenting the results requires significant improvement. I also have major reservations about some of the analyses. Therefore, I ask the authors to address the following comments:

Comments to manuscript:

1.      The sentence in lines 99-100: “The novel C1-C5 complexes were synthesized and characterized by elemental analyses, UV-Vis, IR, 1H NMR and ESI-MS spectroscopy (Figure 1).” suggests that all these spectra (UV-Vis, IR, 1H NMR and ESI-MS) are on Figure 1, which is not true. I  imply to remove reference Figure 1.

2.      The Figure 2 is low resolution. I suggest separating all parts and creating a, b, c, d, sections, each for a different complex. Moreover, please put the equation of the line on the graphs. Also, I suggest cutting the scale of UV-Vis spectra (Figure 2, axis x) and adding a color legend.  

3.      The authors use the name DNA ambiguously in the text. Sometimes it is "DNA" and sometimes it is "CT-DNA". Please standardize the notation. I recommend using CT-DNA , which is much more correct because this material was used in the research.

4.      The authors should briefly explain how exactly the studies on interactions with ct-DNA were carried out using UV-Vis, since studied compounds absorbed light in the examined range.

5.      Figure 3 would look better if the graphs were separated as sections and were bigger. Please also add the equation of the line and the legend.

6.      Figure 4 requires the same changes as Figures 2 and 3.

7.      I suggest separating Figures 5 and 6 on sections a, b, c, …. for each compound.

8.      I suggest separating Figures 5 and 6 into sections a, b, c and loading better resolution and larger plots.

9.      I recommend extending the discussion and comparing your results with literature data regarding the interaction with HSA and CT-DNA.

10.  The Authors should add more information about more information on the origin and composition of the buffers used and ionic strength.

11.  Were fluorescence spectroscopy measurements, in HAS interaction study, carried out without inner-filter effect correction? The use of this correction may affect the obtained values of the KSV constants. Please give some comments or more details of analyses, because it is not clear.  

12.  HSA has multiple binding sites. The Authors should explain where exactly in the protein the heteronuclear complexes bind. I recommend performing fluorescence tests with markers such as phenylbutazone, ibuprofen, or dansyl-glycine, and dansyl-phenylalanine. At the same time, or instead, theoretical considerations can be carried out using model docking.

13.  The Authors should add more details about spectroscopic measurements: kind of UV-Vis lamp, spectra resolution, and number of accumulation. What was the sample complexes concentration in UV-Vis spectra with CT-DNA interaction research? What was the concentration of EtBr/CT-DNA concentration in the competition binding study?

14.  Please check all text carefully, and remove editorial flaws, for example:

- line 45 – “downregulated” should be “down-regulated”;

- line 167 –  “KSV” missed down index;

- line 221 – remove not necessary dot;

- line 245 – requires down index.

Comments to supplementary material:

1.      The figures do not have captions eg.: Figure S1 etc. The authors refer to the figures in the supplementary material in the main text, but they are not properly marked there (in the same way).

2.      It is not clear what unit is on Y axis on UV-Vis spectra. Is it absorbance or maybe absorption? The absorbance should not exceed a value greater than 1. The experimental parameters, concentration, optical path length, etc. should be matched so that the spectrum will be reliable.

3.      The UV-Vis spectra of [ZnCl2(terpytBu)] require other color than yellow. Moreover, the signal locations should also be listed below the spectrum as (X,Y). This would improve the readability of the drawing. These values often overlap in the current form they are written, making reading difficult.

4.      The Y scale of UV-Vis spectra of  [{trans-PtCl(NH3)2(μ-pyrazine)ZnCl(terpytBu)}](ClO4)2 should be strat at 0.0.

5.      I recommend increasing the font size on NMR and IR spectra (peak captions).

6.      I suggest the load the better-quality FT-IR spectra. Current figures are hard to read without a hiper-zoom.

7.      In my opinion, the description of the electronic absorption and fluorescence spectroscopic methods used to determine the Kb and Ksv constants should be moved to the manuscript in the Material and Methods chapter.  

Comments on the Quality of English Language

The article is written in clear and understandable English, although with specific scientific phrases.

Author Response

Response to Comments from Reviewer: 1

Comments to the Author: The cancer is still one of the main causes of death in humans. Therefore, it is so important to searching new diagnostic and therapeutic agents. One of the strategies is finding metal ion complexes. The discovery of cisplatin compounds sets a new direction for research of new anticancer therapies. Although cisplatin is currently used drug in cancer therapy it is severely limited by toxic side effects. For this reason, the search for new metal-dependent complex compounds that have anticancer properties and do not have side effects is an important issue nowadays. The authors of the manuscript: “Modes of interactions with DNA/BSA biomolecules and comparative cytotoxic studies of newly synthesized mononuclear zinc(II) and heteronuclear platinum(II)/zinc(II) complexes toward colorectal cancer cells” present research results on new ruthenium and zinc complexes, including heteronuclear complexes, which is a new approach. The article has a structure typical of research work. The introduction part is short, and quite well-written but could be extended. The work has great potential because it is an interdisciplinary approach. The authors present the history of compounds from synthesis through studies of interactions with macromolecules and biological evaluation. This is the biggest advantage of this work. However, the way of presenting the results requires significant improvement. I also have major reservations about some of the analyses. Therefore, I ask the authors to address the following comments:

Response: Authors are very grateful to the reviewer for his positive and encouraging comments.

Comments to the Author: 1. The sentence in lines 99-100: “The novel C1-C5 complexes were synthesized and characterized by elemental analyses, UV-Vis, IR, 1H NMR and ESI-MS spectroscopy (Figure 1).” suggests that all these spectra (UV-Vis, IR, 1H NMR and ESI-MS) are on Figure 1, which is not true. I imply to remove reference Figure 1.

Response: We highly appreciate the very useful suggestion of the reviewer. We removed the referenced Figure 1.

Comments to the Author:   2. The Figure 2 is low resolution. I suggest separating all parts and creating a, b, c, d, sections, each for a different complex. Moreover, please put the equation of the line on the graphs. Also, I suggest cutting the scale of UV-Vis spectra (Figure 2, axis x) and adding a color legend. 

Response: We highly appreciate the very useful suggestion of the reviewer. We corrected as was suggested.

Comments to the Author:   3. The authors use the name DNA ambiguously in the text. Sometimes it is "DNA" and sometimes it is "CT-DNA". Please standardize the notation. I recommend using CT-DNA , which is much more correct because this material was used in the research.

Response: We highly appreciate the very useful suggestion of the reviewer. We standardized the notation, the CT-DNA is placed instead of DNA as was suggested.

Comments to the Author:  4. The authors should briefly explain how exactly the studies on interactions with ct-DNA were carried out using UV-Vis, since studied compounds absorbed light in the examined range.

Response: It was found that the absorbance properties of complexes C4 and C5 coincide with the range studied for the interaction with the CT DNA molecule. Therefore, the measured absorbance values of the CT-DNA solution were corrected and attributed to the absorbance of the respective complexes. The corrected values are listed in Table 1.

Comments to the Author:  5. Figure 3 would look better if the graphs were separated as sections and were bigger. Please also add the equation of the line and the legend.

Response: Thanks for valuable suggestion. It has been corrected as suggested.

Comments to the Author:  6. Figure 4 requires the same changes as Figures 2 and 3.

Response: Thanks for valuable suggestion. It has been corrected as suggested.

Comments to the Author:  7. I suggest separating Figures 5 and 6 on sections a, b, c, …. for each compound.

Response: Thanks for valuable suggestion. We separated Figures 5 and 6 on section a, b, c, d, e for each complex.

Comments to the Author:  8. I suggest separating Figures 5 and 6 into sections a, b, c and loading better resolution and larger plots.

Response: Thanks for valuable suggestion. We separated and loaded better resolution for Figures 5 and 6.

Comments to the Author:  9. I recommend extending the discussion and comparing your results with literature data regarding the interaction with HSA and CT-DNA.

Response: Thanks for valuable suggestion. We extended the discussion and compared results with literature data regarding the interaction of dinuclear complexes.with HSA and CT-DNA.

Comments to the Author:  10. The Authors should add more information about more information on the origin and composition of the buffers used and ionic strength.

Response: Thanks for valuable suggestion. We adedd more information about origin and composition of buffers and ionic strength.

Comments to the Author:  11. Were fluorescence spectroscopy measurements, in HAS interaction study, carried out without inner-filter effect correction? The use of this correction may affect the obtained values of the KSV constants. Please give some comments or more details of analyses, because it is not clear. 

Response: In the test area where HSA fluoresces, the complexes do not exhibit fluorescence, therefore it was not possible to apply a correction for the inner-filter effect.

Comments to the Author:  12. HSA has multiple binding sites. The Authors should explain where exactly in the protein the heteronuclear complexes bind. I recommend performing fluorescence tests with markers such as phenylbutazone, ibuprofen, or dansyl-glycine, and dansyl-phenylalanine. At the same time, or instead, theoretical considerations can be carried out using model docking.

Response: Thanks for valuable suggestion. We performed molecular docking and explained in manuscript where exactly in the HSA the heteronuclear complexes bind.

Comments to the Author:  13. The Authors should add more details about spectroscopic measurements: kind of UV-Vis lamp, spectra resolution, and number of accumulation. What was the sample complexes concentration in UV-Vis spectra with CT-DNA interaction research? What was the concentration of EtBr/CT-DNA concentration in the competition binding study?

Response: Thanks for valuable suggestion. We have made improvements to the manuscript incorporating all suggestions.

Comments to the Author:  14. Please check all text carefully, and remove editorial flaws, for example:

- line 45 – “downregulated” should be “down-regulated”;

- line 167 –  “KSV” missed down index;

- line 221 – remove not necessary dot;

- line 245 – requires down index.

Response: Thanks for valuable suggestion. We checked the text and corrected all typographical errors.

Comments to supplementary material:

Comments to the Author:  1. The figures do not have captions eg.: Figure S1 etc. The authors refer to the figures in the supplementary material in the main text, but they are not properly marked there (in the same way).

Response: Thanks for valuable suggestion. We added captions to the figures in Supplementary material

Comments to the Author:  2. It is not clear what unit is on Y axis on UV-Vis spectra. Is it absorbance or maybe absorption? The absorbance should not exceed a value greater than 1. The experimental parameters, concentration, optical path length, etc. should be matched so that the spectrum will be reliable.

Response: Thanks for valuable suggestion. We added captions to the figures in Supplementary material

Comments to the Author:  3. The UV-Vis spectra of [ZnCl2(terpytBu)] require other color than yellow. Moreover, the signal locations should also be listed below the spectrum as (X,Y). This would improve the readability of the drawing. These values often overlap in the current form they are written, making reading difficult.

Response: Thanks for comments. The UV-Vis are now black color. The signal locations of the peaks are listed below spectra. We recorded spectra every 2 nm and we have got at least 500 values, because of that the values of main peaks are presented.

Comments to the Author:  4. The Y scale of UV-Vis spectra of [{trans-PtCl(NH3)2(μ-pyrazine)ZnCl(terpytBu)}](ClO4)2 should be strat at 0.0.

Response: Thanks for comment. The Y scale UV-Vis now start at 0.0.

Comments to the Author:  5. I recommend increasing the font size on NMR and IR spectra (peak captions).

Response: Thanks for recommendation. We increased the font size as much as the programs have allowed.

Comments to the Author:  6. I suggest the load the better-quality FT-IR spectra. Current figures are hard to read without a hiper-zoom.

Response: Thanks for suggestion. We tried to load the better-quality FT-IR spectra. FT-IR spectra were recorded on two different instruments and persons responsible send us as PDF files the spectra because of that it was hard to read without a hiper-zoom.

Comments to the Author:  7.  In my opinion, the description of the electronic absorption and fluorescence spectroscopic methods used to determine the Kb and Ksv constants should be moved to the manuscript in the Material and Methods chapter. 

Response: Thanks for suggestion. We moved the description in Material and Methods chapter as suggested.

Reviewer 2 Report

Comments and Suggestions for Authors

In the manuscript titled “Modes of interactions with DNA/BSA biomolecules and comparative cytotoxic studies of newly synthesized mononuclear zinc(II) and heteronuclear platinum(II)/zinc(II) complexes toward colorectal cancer cells” the authors designed 5 compounds based on mononuclear zinc(II) and heteronuclear platinum(II)/zinc(II) complexes and successfully synthesized them. The work is novel and sounds interesting, but needs many corrections before it can be considered for publication. Please give proper corrections or comments on the following points:

1.     In the experiments listed in Table 1, binding constant (kb) and Stern-Volume constant (KSV), the authors need to perform the tests on reference compounds for comparison with the obtained results for the synthesized compounds.

2.     Why the authors did not perform 13C NMR for the confirmation of the chemical identities of the obtained compounds?

3.     The authors concluded that the synthesized compounds demonstrated significant cytotoxic effects, however, none of the compounds exhibited a single-digit micromolar inhibition against any of the tested cancer cell lines. Would you please explain this?

4.     In section 2.4. the authors need to make the following:

a.       Docking validation.

b.       Giving the result for the docking of the original ligand (co-crystalized with the selected proteins) and comparing it to the docked compounds in terms of pose and affinity.

5.     In section 3.3.3. please give the detailed results for compound C1 and the elemental analysis for all compounds.

6.     In section 3.6.1. why there is no antibiotic added to culture media? And what is temperature and relative humidity?

7.     The size of the figures and scheme 1 need to be enlarged with good resolutions.

8.     The manuscript needs to be extensively revised for language clarity and grammar mistakes.

Examples:

Line 32. Add (of) after “binding properties”

Line 34. Add (and) after “C4”

Line 37. Value should be values

Line 39. Complexes should be complex

Comments on the Quality of English Language

The manuscript need to be extensively revised for language clarity 

Author Response

Response to Comments from Reviewer: 2

Comments to the Author: In the manuscript titled “Modes of interactions with DNA/BSA biomolecules and comparative cytotoxic studies of newly synthesized mononuclear zinc(II) and heteronuclear platinum(II)/zinc(II) complexes toward colorectal cancer cells” the authors designed 5 compounds based on mononuclear zinc(II) and heteronuclear platinum(II)/zinc(II) complexes and successfully synthesized them. The work is novel and sounds interesting, but needs many corrections before it can be considered for publication. Please give proper corrections or comments on the following points:

 Response: Authors highly appreciate the reviewer’s positive and valuable comments.

Comments to the Author: 1.   In the experiments listed in Table 1, binding constant (Kb) and Stern-Volume constant (KSV), the authors need to perform the tests on reference compounds for comparison with the obtained results for the synthesized compounds.

Response: When comparing the constants of the heteronuclear complexes C2-C5 with those of the complexes from which the newly synthesized complexes were derived (transplatin, cisplatin, and the C1 complex), it is observed that complexes C2, C4, and C5 (as per Table 1) exhibit significantly enhanced constant values compared to the initial complexes. Nevertheless, complex C2 maintains values akin to the starting complexes. These findings suggest that the construction of heteronuclear complexes contributes to improved reactivity, particularly in the instances of complexes C2, C4, and C5, while complex C2 demonstrates reactivity similar to that of the starting complexes.

Comments to the Author: 2. Why the authors did not perform 13C NMR for the confirmation of the chemical identities of the obtained compounds?

Response: Due to low solubility of the complexes, their concentrations in prepared samples were not good enough for 13C NMR measurements on our 200 MHz instrument.

Comments to the Author: 3. The authors concluded that the synthesized compounds demonstrated significant cytotoxic effects, however, none of the compounds exhibited a single-digit micromolar inhibition against any of the tested cancer cell lines. Would you please explain this?

Response: Thanks for comments. We meant that complex C1 demonstrated pronounced selectivity towards cancer cells over non-cancerous mMSC activity than others complexes, but how we expressed it was wrong, and we corrected our conclusion.

Comments to the Author: 4.   In section 2.4. the authors need to make the following:

  1. Docking validation.

  1. Giving the result for the docking of the original ligand (co-crystalized with the selected proteins) and comparing it to the docked compounds in terms of pose and affinity.

Response: Thanks for suggestions. We made docking validation and gave results for docking of the original ligand and compared it to the docked compounds in terms of pose and affinity.

Comments to the Author:   5. In section 3.3.3. please give the detailed results for compound C1 and the elemental analysis for all compounds.

Response: Thanks to comments. We gave the detailed results for compound C1 in Section 3.3. but we agreed with the reviewer and moved it in Section 3.3.3. The elemental analysis for all complexes already has been provided.

Comments to the Author:   6. In section 3.6.1. why there is no antibiotic added to culture media? And what is temperature and relative humidity?

Response: We appreciated the questions. We have added in media 100 IU/ml penicillin and 100 mg/mL streptomycin antibiotics (Invitrogen, USA). Samples were incubated at 37 °C in in a humidified atmosphere of 5% carbon dioxide (CO2). Additional explanations now are provided in Section 3.6.1.

Comments to the Author:  7. The size of the figures and scheme 1 need to be enlarged with good resolutions

Response: Thanks for suggestion. We enlarged the size figures and schemes with good resolution as was suggested.

Comments to the Author:  8. The manuscript needs to be extensively revised for language clarity and grammar mistakes.

Examples:

Line 32. Add (of) after “binding properties”

Line 34. Add (and) after “C4”

Line 37. Value should be values

Line 39. Complexes should be complex

Response: Thanks for comments. The manuscript was extensively revised for language clarity and grammar mistakes. All changes are highlighted in red.

Round 2

Reviewer 1 Report

Comments and Suggestions for Authors

The changes introduced contributed to the increase in the quality of the article. I believe that in its final form, it is recommended for publication International Journal of Molecular Science.

Comments on the Quality of English Language

English language required minor editing. 

Reviewer 2 Report

Comments and Suggestions for Authors

The authors have modified the manuscript significantly and I can now recommend it for publication.